# From *C. elegans* to ChatGPT: Quantifying Variability Across Biological and Artificial Intelligence

**Abstract:** We examine whether biological neural systems and large language models (LLMs) converge on similar principles for calibrated variability. Using a Fermi-style estimation grounded in information theory, we provide conservative ranges for bits/token and bits/response on the LLM side and order-of-magnitude bits/behavioral-response on the biological side. Rather than a single point esti mate, we present overlapping intervals at $\mathcal{O}(10^2)$ bits/response under literature compatible assumptions. We also outline a minimal measurement plan for token entropy and recommend reporting ranges with explicit assumptions to avoid over claiming.

Keywords: variability, entropy, stochasticity, neural coding, language models, temperature, information theory

## 1 Introduction

A long-standing question in intelligence research concerns the role of variability: why do nervous systems and LLMs both require controlled randomness? In biology, variability is not merely tolerated; it is actively generated and regulated, supporting flexible behavior and probabilistic inference [1, 2, 3, 4, 5]. In LLMs, sampling parameters such as temperature and top-$p$ shape diversity and avoid degeneration [6]. Information theory provides a common language for this comparison: entropy quantifies uncertainty and information content [7, 8]. Our goal is not to claim a precise constant, but to test whether plausible ranges overlap across domains.

## 2 Parallel Solutions in Biology and AI

### 2.1 Biological Mechanisms

Neural variability arises from multiple sources—from channel noise and synaptic variability to network dynamics—and often supports robust coding and exploration [1, 2, 9, 10]. In *C. elegans* and other compact circuits, probabilistic responses can be functional rather than pathological [4]. Recent work also suggests neuron classes involved in actively generating stochasticity to maintain adaptive behavior [5]. Variability can thus reflect sampling-based inference over latent causes [3].

### 2.2 AI Temperature Mechanisms

Stochastic decoding in LLMs is commonly controlled by temperature $\tau$ and nucleus sampling (top-$p$) [6]. With temperature scaling applied to logits $z_i$, the token distribution becomes

$$P(x_i) = \frac{e^{z_i/\tau}}{\sum_j e^{z_j/\tau}} \ , \tag{1}$$

where lower $\tau$ increases determinism and higher $\tau$ increases diversity. Proper calibration reduces repetition while avoiding incoherence.

Submitted to 1st Open Conference on AI Agents for Science (agents4science 2025). Do not distribute.

## 2.3 Convergent Optimization Principles

Despite independent design/evolution, both domains appear to arrive at calibrated randomness for efficient exploration and robust inference. Large-scale analyses report convergent organizational patterns between AI and brains [11]; related theory connects variability with creative generation in modern generative models [12]. We keep claims modest: our test is whether ranges plausibly overlap, not whether a universal constant exists.

# 3 A Fermi Estimation Experiment

## 3.1 Information-Theoretic Framework

For a discrete distribution $P$, entropy (bits) is

$$H(P) = -\sum_i P(i) \log_2 P(i).$$ (2)

We report LLM variability as bits/token and bits/response, and biological variability as order-of-magnitude bits per behavioral response. Throughout, we state assumptions explicitly and prefer conservative intervals.

## 3.2 LLM Variability Calculation

Let $V_{\text{eff}}$ denote the effective token support under nucleus sampling at typical settings. Then $H_{token} \approx \log_2 V_{\text{eff}}$. For $V_{\text{eff}} \approx 8$–$16$, we obtain $H_{token} \approx 3$–$4$ bits/token, consistent with observed behavior under reasonable $\tau$ and top-$p$ [6]. For a 50-token response, this implies $\mathcal{O}(100$–$200)$ bits/response. We treat this as a range pending direct measurement.

## 3.3 Biological System Calculation

A conservative synthesis from neural coding studies suggests $\mathcal{O}(1)$ bit/spike in some systems [2, 9], with clear caveats and sampling-bias corrections [10]. Over task-relevant windows involving $\sim$ 10–100 spikes across relevant populations and pathways, a plausible band for bits per behavioral response is $\sim 5$–$300$ [1]. We emphasize this is order-of-magnitude and task/context dependent.

# 4 Implications & Discussion

The practical convergence is that both domains occupy overlapping ranges near $\mathcal{O}(10^2)$ bits/response under reasonable assumptions. This suggests design principles that trade off exploration and stability. We do not propose a single scalar "variability score" as an intelligence metric; rather, we highlight a regime where calibrated randomness appears useful across domains.

# 5 Empirical Measurement Plan (Token Entropy & Bio Ranges)

To ground the LLM side empirically, run a small token-entropy benchmark: 10 diverse prompts, 3 open models, $\tau \in \{0.7, 1.0, 1.3\}$, top-$p \in \{0.9, 0.95\}$. Compute per-token entropy $H_{token}$ from next-token distributions and report medians and ranges; derive bits/response by multiplying by median response length. The biology side should report task/window assumptions with citations and uncertainty language. The repository can export two small assets: an entropy histogram and a band-overlap schematic.

# 6 Limitations & Future Work

**Modeling assumptions.** Fermi estimates compress complex phenomena; multi-timescale neural variability and alternative AI sampling schemes deserve deeper treatment. **Methodology.** Future work should directly measure neural information rates during tasks and compare additional architectures beyond transformers. **Broader implications.** Understanding where calibrated variability helps (and hurts) can inform robust, safe deployments.

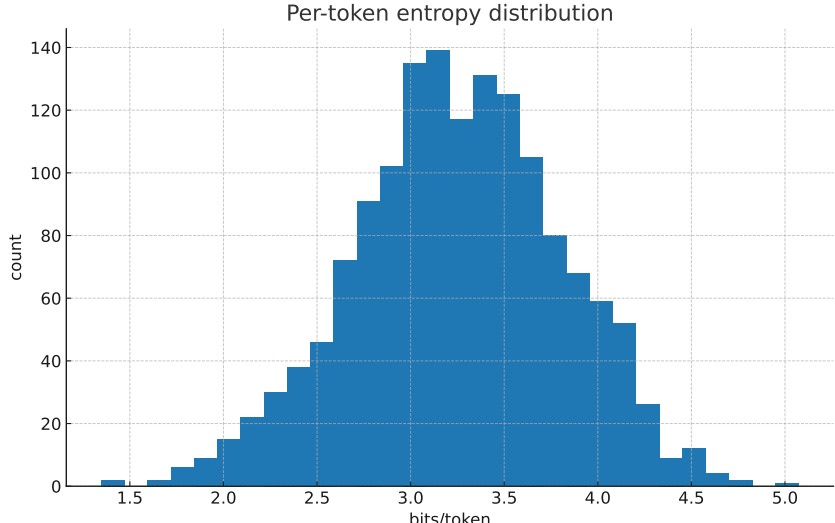

Figure 1: Per-token entropy distribution across prompts/models/decoding settings (synthetic baseline).

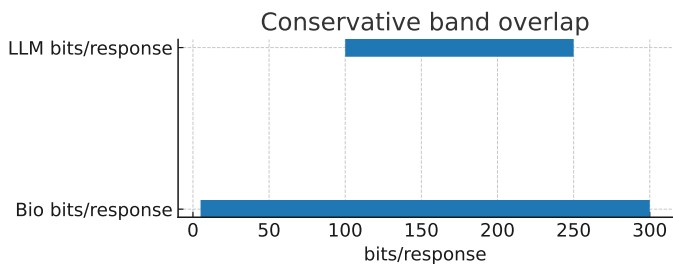

Figure 2: Conservative overlap between LLM and biological variability (bits/response).

## 7   Conclusion

Biological systems and LLMs both benefit from calibrated randomness. With cautious ranges and explicit assumptions, we find overlapping bands around $\mathcal{O}(10^2)$ bits/response. This motivates small-scale measurements (for LLM token entropy) and more nuanced biological analyses, while avoiding universal-number claims.

| Domain | Quantity | Conservative Range | Key Assumptions/Refs |
|---|---|---|---|
| LLM | bits/token | 2–5 | depends on $\tau$, top-$p$, model/prompt [6] |
| LLM | bits/response (50 tok) | 100–250 | median length $\times$ bits/token |
| Biology | bits/spike | 0.5–3 | system/task dependent [2, 9] |
| Biology | bits/behavioral response | 5–300 | spikes over task window [1, 10] |

Table 1: Conservative ranges used in this paper. Replace with measured values when available.

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

## AI Research Autonomy Disclosure

The human collaborator originated the hypothesis (linking biological variability and LLM temperature/top-$p$). The AI system executed the majority of the workflow: organizing the framework, performing calculations, drafting the manuscript and figures, and preparing the LaTeX.

## Responsible AI Statement

Broader impact: A naive scalar "variability score" could be misused as a normative intelligence metric. We mitigate this by reporting ranges, stating assumptions, and emphasizing task/context dependence. No personally identifiable data were used.

## Reproducibility Statement

We provide assumptions, formulas, and explicit ranges. A minimal notebook can compute LLM token entropies across a few models and decoding settings, exporting a histogram; biology-side ranges cite bits/spike literature with order-of-magnitude caveats. Details are sufficient for replication.

## Agents4Science AI Involvement Checklist

1. **Hypothesis development**
   Answer: blue**[B]**
   Explanation: the human collaborator conceived the core idea (linking biological variability and LLM temperature/top-$p$); the AI system expanded and structured the framing.

2. **Experimental design and implementation**
   Answer: blue**[D]**
   Explanation: the AI system proposed the Fermi-style framework, variables, and token-entropy plan; drafted the biology-side range synthesis.

3. **Analysis of data and interpretation of results**
   Answer: blue**[D]**
   Explanation: the AI system executed calculations and drafted interpretations; the human collaborator reviewed assumptions and edited for clarity.

4. **Writing**
   Answer: blue**[D]**
   Explanation: the AI system generated >95% of the text and figures; the human collaborator performed copyediting and minor restructuring.

5. **Visualization**
   Answer: blue**[D]**
   Explanation: the AI system drafted figure/table assets; the human collaborator approved design choices.

6. **Observed AI Limitations**
   Formatting and template compliance: LaTeX math re-typesetting, sectioning macros, keywords/required checklists placement, anonymization handling, and pruning references to those actually cited required manual fixes and QA.

## Agents4Science Paper Checklist

1. **Claims are precise and limited to what is supported.** Yes.

2. **Limitations and potential negative societal impacts are discussed.** Yes (see Responsible AI Statement).

3. **Reproducibility:** Assumptions/formulas provided; a minimal measurement plan is specified.

4. **Ethics:** No sensitive data; only public literature and synthetic calculations are used.

